# Biological Control Activity of Plant Growth Promoting Rhizobacteria *Burkholderia contaminans* AY001 against Tomato Fusarium Wilt and Bacterial Speck Diseases

**DOI:** 10.3390/biology11040619

**Published:** 2022-04-18

**Authors:** A Yeong Heo, Young Mo Koo, Hyong Woo Choi

**Affiliations:** 1Department of Plant Medicals, College of Life Sciences and Biotechnology, Andong National University, Andong 36729, Korea; ayeong412@korea.kr (A.Y.H.); rndudah130@student.anu.ac.kr (Y.M.K.); 2Division of Forest Insect Pests & Diseases, National Institute of Forest Science, Seoul 02455, Korea

**Keywords:** *Burkholderia contaminans*, PGPR, tomato, Induced Systemic Resistance, biocontrol

## Abstract

**Simple Summary:**

*Burkholderia contaminans* belongs to *B*. *cepacia* complex (Bcc), those of which are found in various environmental conditions. In this study, a novel strain AY001 of *B*. *contaminans* (AY001) was identified from the rhizosphere soil sample. AY001 showed (i) various plant growth-promoting rhizobacteria (PGPR)-related traits, (ii) antagonistic activity against different plant pathogenic fungi, (iii) suppressive activity against tomato Fusarium wilt disease, (iv) induced systemic acquired resistance (ISR)-triggering activity, and (v) production of various antimicrobial and plant immune-inducing secondary metabolites. These results suggest that AY001 is, indeed, a successful PGPR, and it can be practically used in tomato cultivation to alleviate biotic and abiotic stresses. However, further safety studies on the use of AY001 will be needed to ensure its safe use in the Agricultural system.

**Abstract:**

Plant growth promoting rhizobacteria (PGPR) is not only enhancing plant growth, but also inducing resistance against a broad range of pathogens, thus providing effective strategies to substitute chemical products. In this study, *Burkholderia contaminans* AY001 (AY001) is isolated based on its broad-spectrum antifungal activity. AY001 not only inhibited fungal pathogen growth in dual culture and culture filtrate assays, but also showed various PGPR traits, such as nitrogen fixation, phosphate solubilization, extracellular protease production, zinc solubilization and indole-3-acetic acid (IAA) biosynthesis activities. Indeed, AY001 treatment significantly enhanced growth of tomato plants and enhanced resistance against two distinct pathogens, *F*. *oxysporum* f.sp. *lycopersici* and *Pseudomonas syringae* pv. *tomato*. Real-time qPCR analyses revealed that AY001 treatment induced jasmonic acid/ethylene-dependent defense-related gene expression, suggesting its Induced Systemic Resistance (ISR)-eliciting activity. Gas chromatography–mass spectrometry (GC-MS) analysis of culture filtrate of AY001 revealed production of antimicrobial compounds, including di(2-ethylhexyl) phthalate and pyrrolo [1,2-a]pyrazine-1,4-dione, hexahydro-3-(phenylmethyl). Taken together, our newly isolated AY001 showed promising PGPR and ISR activities in tomato plants, suggesting its potential use as a biofertilizer and biocontrol agent.

## 1. Introduction

Tomato (*Solanum lycopersicum* L.) is the second most grown vegetable in the world after potato. It accounts for 16% (180 million metric tons) of the world’s vegetable primary production in 2019 (www.fao.org/faostat, accessed on 1 July 2021). Nearly 200 different species of pathogens, including fungi, bacteria, viruses, and others, are known to be able to cause disease in tomato plants [1]. One of the major problems in tomato cultivation is Fusarium wilt disease, caused by *Fusarium oxysporum* f. sp. *lycopersici* (*FOL*). Germ tube and mycelium of *FOL* are able to directly invade the root tip, or enter to root through a wound or lateral root [2]. Then *FOL* reaches the xylem vessels and mycelium grows mostly upward along the stem and crown, thus inducing gradual wilting symptoms and eventual death [3]. Once the soil is contaminated by *FOL*, it is very difficult to control by chemical fungicides as it can survive for long periods [4]. There is an increasing effort to provide a new strategy to control soil-borne fungus, like *FOL*, using biological control agents [5,6,7].

The use of microorganism as a biological control agent is receiving more and more attention as (i) it is a sustainable and environmentally friendly way to alternate chemical pesticides, (ii) it can demonstrate good adaptability to the environment depending on its lifestyle, and (iii) it has versatile working mechanisms and synergistic interactions with plants, unlike chemicals [8]. Especially, beneficial soil bacteria that inhabit the roots or rhizosphere and promote plant growth and development are collectively referred to as plant growth-promoting rhizobacteria (PGPR) [9,10]. PGPR are able to promote plant growth either directly (mineral solubilization, nitrogen fixation, and phytohormone production, such as auxin, gibberellin, and cytokines) or indirectly (production of phenazine, hydrogen cyanide, and siderophore) [10]. In addition, PGPR can trigger Induced Systemic Resistance (ISR) in plants, thereby enhancing the resistance against various pathogens [11]. ISR is known to be regulated by jasmonic acid (JA) and ethylene (ET) signaling pathways [12]. Induction of ISR was demonstrated by different species of PGPR, such as *Pseudomonas* spp., *Bacillus* spp., and *Burkholderia* spp. which reduced the incidence or severity of different diseases in various host plants [13,14,15].

*Burkholderia* spp. have been isolated from various ecological niches, including plants, soils, rhizosphere, animals, and humans, and can establish relationships with a wide range of plants [16,17,18], suggesting its ability to adapt to various environmental conditions. Most *Burkholderia* spp. isolated from the soil are associated with plants, and some of them showed remarkable PGPR and ISR-inducing activities in different host plants [19]. However, several *Burkholderia* spp., such as *B*. *cenocepacia*, *B*. *multivorans,* and *B*. *dolosa*, are suggested as opportunistic pathogens in humans, which can cause serious respiratory infections in cystic fibrosis (CF) and immunocompromised patients [20]. Thus it is very important to understand the virulence mechanism of *Burkholderia* spp. in humans and/or animals. However, its virulence mechanism is not fully understood yet, due to its high genomic diversity even within the same type of *Burkholderia* spp. [19,20]. Thus careful safety studies will be needed before the use of *Burkholderia* spp. in the Agriculture system.

In this study, we have newly isolated and characterized *Burkholderia contaminans* strain AY001 (AY001) from soil sample. AY001 showed distinct antagonistic effects on different plant pathogenic fungi and various PGPR-related traits in vitro. In tomato plants, AY001 treatment (i) enhanced growth, (ii) reduced disease severity of Fusarium wilt by direct antagonistic mechanisms, and (iii) enhanced resistance against *Pseudomonas syringae* pv. *tomato* by inducing ISR. Gas chromatography-mass spectrometry (GC-MS) analysis of hexane and ethyl acetate extracts of culture filtrate of AY001 revealed possible mechanisms of action of its antagonistic activity. Together, AY001 demonstrated great potential as a biofertilizer and biological control agent.

## 2. Materials and Methods

### 2.1. Isolation and Screening of Antagonistic Bacteria

The rhizosphere soils of various weed plants were sampled from Andong, South Korea. One gram of soil samples was suspended with 10 mL sterilized distilled water and vortexed for 1 min. To isolate rhizosphere bacteria, serial dilutions of soil suspensions were grown on nutrient agar media (NA, KisanBio, Seoul, Korea). Unknown bacterial strains were isolated and screened for their antagonistic activity against *Fusarium oxysporum* f. sp. *lycopersici* strain KACC 40038 (*FOL*) on potato dextrose agar (PDA, KisanBio, Korea) by dual culture assay [7]. Among 155 strains screened, AY001 showed the strongest inhibition activity on the growth of *FOL*, thus selected for further study.

### 2.2. Molecular Identification of AY001

Total genomic DNA of AY001 was extracted using HiGene Genomic DNA Prep Kit (BIOFACT, Daejeon, Korea) according to the manufacturer’s instructions. Molecular identification of AY001 was performed by sequencing the amplified 16S rRNA region using primers 27F (5′-AGTTTGATCCTGGCTCAG-3′) and 1492R (5′-GTTACCTTGTTACGACTT-3′). PCR amplification was performed in a 30 μL reaction mixture containing 15 μL of 2X Taq PCR Pre-Mix (Solgent, Daejeon, Korea), 1 μL of primers mix, and 1 μL of gDNA template. PCR amplification was carried out using a thermal cycler (Multigene Gradient, Labnet, Edison, NJ, USA) by following amplification conditions: initial denaturation at 95 °C for 5 min, 33 cycles of 94 °C for 30 s, 58 °C for 30 s, 72 °C for 40 s, and final extension at 72 °C for 10 min. PCR product was purified, sequenced (Solgent, Daejeon, Korea), and analyzed with NCBI’s GenBank sequence database (http://www.ncbi.nlm.nih.gov, accessed on 1 March 2020) to identify the closest species relatives. The phylogenetic tree was constructed by MEGA-X software using the Maximum Likelihood method [21].

### 2.3. Dual Culture and Culture Filtrate Assays

Dual culture assay was performed to confirm the antagonistic effect of AY001 against *FOL*, *F. avenaceum*, *F. solani*, *C. acutatum*, *P. capsici*, and *S. sclerotiorum* on PDA media as previously described with minor modifications [7]. Due to the different growth rates, pathogen growth inhibition rate was measured at 5 days post-inoculation (dpi) for *S. sclerotiorum*, 10 dpi for *FOL,* and *F. avenaceum*, and 14 dpi for *F. solani*, *C. acutatum*, *P. capsici*. Pathogen growth inhibition rate (%) was calculated according to the following equation: (100-(diameter of the pathogen in the presence of AY001/diameter of the pathogen in the absence of AY001 × 100)).

Antifungal activity of culture filtrate (CF) of AY001 was tested as previously described [7]. Briefly, AY001 was cultured in nutrient broth (NB) media at 25 °C for 2 days, centrifuged at 13,000 rpm for 10 min, and the supernatant was filtered with 0.22 μm filters. For culture filtrate assay, different fungal pathogens were grown on PDA media supplemented with either 0 or 10% CF of AY001, and inhibition rate was measured using ImageJ (NIH).

### 2.4. Zinc and Phosphate Solubilization

Zinc solubilization activity was tested as previously described [22]. Briefly, AY001 was grown on tris-minimal salt medium containing 10.0 g D-glucose, 6.06 g Tris-HCl, 4.68 g NaCl, 1.49 g KCl, 1.07 g NH_4_Cl, 0.43 g Na_2_SO_4_, 0.2 g MgCl_2_ 2H_2_O, 30 mg CaCl_2_ H_2_O and 15.0 g agar/L (pH7.0). To check the zinc solubilization activity 0.1% ZnO (1.244 g/L) was added to the medium. Phosphate solubilization activity was tested on National Botanical Research Institute’s phosphate growth (NBRIP) medium containing 10.0 g D-glucose, 5.0 g Ca_3_(PO_4_)_2_, 5.0 g MgCl_2_ 6H_2_O, 0.25 g MgSO_4_ 7H_2_O, 0.2 g KCl, 0.1 g (NH_4_)_2_SO_4_ and 15.0 g agar/L (pH7.0) [23]. AY001 was grown on tris-minimal salt and NBRIP medium and incubated at 28 °C for up to 10 days. Solubilization Index (SI) was calculated according to the following formula (total diameter (colony diameter + halo zone diameter)/colony diameter).

### 2.5. Protease Activity

Protease activity of AY001 was tested on skim milk agar medium. To prepare the medium, 400 mL of skim milk solution (10.0 g skim milk powder in 400 mL sterile deionized water (SDW)) and 600 mL of yeast agar medium (1.0 g yeast extract and 30.0 g agar in 600 mL SDW) were autoclaved, separately. Each solution was cooled to 50 °C, then mixed and poured into plates. AY001 was cultivated on Skim milk agar medium and incubated at 28 °C for 10 days. Hydrolysis Index (HI) was calculated according to the following formula (total diameter (colony diameter + halo zone diameter)/colony diameter).

### 2.6. Siderophore Production

Chrom azurol S (CAS) assay was used to check the siderophore production activity of AY001 as previously described [24]. Briefly, CAS agar plates were prepared by mixing CAS, hexadecyl-trimethyl-ammonium (HDTMA), and FeCl_3_ dye. For CAS dye, 60.5 mg CAS (Sigma, Ronkonkoma, NY, USA) was dissolved in 50 mL of SDW and mixed with 10 mL of 1 mM FeCl_3_ and 40 mL of HDTMA solution 72.9 mg/40 mL SDW), then autoclaved and stored under the dark condition. The CAS solution (100 mL) was mixed with NA media (900 mL), and poured into plates. After that, AY001 was inoculated and incubated at 28 °C for 5 days. Siderophore Index (SI) was calculated according to the following formula (total diameter (colony diameter + halo zone diameter)/colony diameter).

### 2.7. Ammonia (NH_3_) Production

Ammonia production activity was examined as previously described [25]. Peptone broth medium (peptone 10 g, NaCl 5.0 g, and pH was adjusted to 7.2) was used for ammonia (NH_3_) production assay. After AY001 inoculation into peptone broth media, it was incubated in a shaking incubator with 200 rpm at 28 °C. To monitor NH_3_ production, the culture was sampled and centrifuged every 24 h up to 10 days, then 1 mL of supernatant reacted with 1 mL of Nessler’s reagent (Duksan Science, Seoul, Korea) and made up to 10 mL by addition of ammonia-free SDW. The OD_450 nm_ value was measured to quantify NH_3_ production.

### 2.8. Indole-3-Acetic Acid (IAA) Production

IAA production activity was evaluated in an NB medium with different concentrations of L-tryptophan (L-TRP; 0.2 to 2.0 mg/mL) as previously described [26]. OD_600_ value of the AY001 suspension was adjusted to 1.0 and used as an inoculum. Bacterial suspensions in NB media with different concentrations of L-TRP were incubated in a shaking incubator with 200 rpm at 28 °C. After 24 h, 3 mL of the AY001 culture was centrifuged at 13,000 rpm for 10 min at 4 °C. Then, 2 mL of supernatant was mixed with 4 mL of Salkowski’s reagent (0.5 M FeCl_3_ˑ6H_2_O and 35% perchloric acid), and incubated in the dark for 30 min at room temperature. IAA production was quantified by measuring the OD value at 530 nm using spectrophotometer.

### 2.9. Nitrogen Fixation

Nitrogen fixation activity of AY001 was measured using N-free BAz medium containing 2.0 g azelaic acid, 0.4 g K_2_HPO_4_, 0.4 g KH_2_PO_4_, 0.4 g MgSO_4_ˑ7H_2_O, 0.2 g CaCl_2_, 0.002 g Na_2_MoO_4_ˑH_2_O, 0.01 g FeCl_3_, 0.075 g bromothymol blue/L (pH 5.7) [27]. Bacterial suspensions were incubated in shaking incubator with 200 rpm at 28 °C, and pH was measured for 7 days to evaluate nitrogen fixation level. Bromothymol blue-mediated discoloration of media into blue was also observed during the nitrogen fixation experiment.

### 2.10. Plant Growth Condition and Growth of FOL and AY001

Tomato (*Solanum lycoperiscum* L. cv. Seogwang) plants were grown in 5 × 10 cells plastic tray (5.4 × 2.7 × 4.8 cm each cell) for two weeks on the plant culture rack (JSR, Gongju, Korea) at 25 °C with 14 light/10 dark cycle. *FOL* strain KACC 40038 was used as a pathogen. *FOL* was grown on PDA for 10 days at 28 °C. Then, surface of fungal mycelia was filled with 5 mL of SDW and spores were collected using spreader. The spore concentration of *FOL* was adjusted to 1 × 10^6^ conidia/mL using a hemocytometer (PAUL MARIENFELD SUPERIOR, Lauda-Königshofen, Germany). AY001 was inoculated into NB media and incubated at 28 °C in shaking incubator with 200 rpm for 48 h. Then, the concentration was adjusted to 2 × 10^7^ cfu/mL (OD_600_ = 1) and 2 × 10^5^ cfu/mL (OD_600_ = 0.01).

### 2.11. AY001 Treatment and FOL Inoculation

Two-week-old tomato plants were uprooted and treated with either SDW (non-treated control) or AY001 suspension for 30 min at 24 h before inoculation with *FOL* (see below for details). On next day, roots of tomato plants were again submerged into spore suspension of *FOL* for pathogen inoculation. Then, tomato plants were planted in a new pot (10 × 9 cm) containing commercial horticultural media Baroker (Seoul Bio Co., Ltd., Suncheon, Korea) for observation. Baroker media contained 4, 7, 6, 68, and 15% of zeolite, perlite, vermiculite, cocopeat, and peat moss, respectively. In this experiment six treatments were used: (i) SDW, (ii) low (2 × 10^5^ cfu/mL) and (iii) high 2 × 10^7^ cfu/mL) concentrations of AY001, (iv) *FOL* (1 × 10^6^ conidia/mL), (v) low concentration of AY001 and *FOL* and (vi) high concentration of AY001 and *FOL*. AY001 was pre-treated 24 h before the inoculation with *FOL*. Each experiment was performed with 8 replicates. Experiments were repeated 3 times with similar results.

### 2.12. Root Colonization Assay

Root colonization of AY001 in tomato plants was determined as previously described [28,29]. Two-week-old tomato plants were uprooted and submerged into AY001 suspension (OD_600_ = 1; 2 × 10^7^ cfu/mL) for 30 min, then planted in a new plastic pot (10 × 9 cm). Roots were harvested at 0, 1, 3, 5, 7, 10, and 14 days after AY001 treatment. The harvested roots were surface sterilized by soaking in 1% NaOCl for 30 s and rinsed three times with sterilized water. Then roots were weighed to 0.1 g and grounded in 300 µL of SDW using a silamat S6 (Ivolar Vivodent) and glass beads. Serial dilutions were determined using a dot-plating test on NA medium and colony-forming units (cfu) were counted after 24 h incubation at 28 °C. Four replications were evaluated for each experiment and the experiment was repeated 3 times with similar results.

### 2.13. Pst DC3000 Inoculation and Bacterial Growth in Tomato Plants

To test whether AY001 induces resistance against *Pst* DC000, AY001 was pre-treated either on tomato leaves or on roots. For leaf treatment, AY001 suspension (OD_600_ = 1.0; 2 × 10^7^ cfu/mL) or sterilized tap water (STW) was sprayed on three-week-old tomato leaves at 24 h before inoculation with *Pst* DC3000. For root treatment, three-week-old tomato roots were uprooted and submerged either into AY001 suspension (OD_600_ = 1.0) or STW for 30 min, then planted into a new plastic pot (10 × 9 cm). After 24 h, *Pst* DC3000 was inoculated by the spraying method. For *Pst* DC3000 inoculation, *Pst* DC3000 was cultured in KB medium (MBcell, Seoul, Korea) containing 50 μg/mL rifampicin (Rif) at 28 °C for 24 h. Bacterial cells were collected by centrifugation, resuspended in 10 mM MgCl_2_ solution (1 × 10^6^ cfu/mL) with 0.05% Tween 20, and evenly sprayed over the entire leaves. Each treatment was performed with 8 replicates. To measure the *Pst* DC3000 growth, 6 discs of tomato leaves were collected by using a cork borer (diameter = 6.5 mm). Samples were surface sterilized with 1% NaOCl for 30 s, rinsed three times with SDW, then grounded in 300 µL SDW using a silamat S6 and glass beads. Serial dilutions were placed on a KB agar plate containing 50 μg/mL Rif by the dot-plating method. Experiments were repeated 3 times with similar results.

### 2.14. Real-Time qRT-PCR Analysis of Marker Gene Expression

To test marker gene expression, total RNA was extracted from tomato leaves (100 mg) with APure™ Total RNA Kit (GenomicBase, Seoul, Korea) according to the manufacturer’s instruction. For this experiment, AY001 was treated on the root of tomato plants as described above. One µg Total RNA was heated to 65 °C for 5 min with 50 µM Oligo (dT) 20 primer, then cooled on ice. cDNA was synthesized using an RT Series kit (BioFACT™, Daejeon, Korea) according to manufacturer’s instructions. Real-time qRT-PCR was performed using SYBR Green Real-time PCR Master Mix as suggested by the manufacturer (TOYOBO, Osaka, Japan) with the primers listed in Appendix A on a LineGene9600 Plus (GenomicBase, Korea). Briefly, amplification was performed in a 20 μL reaction mixture containing 10 µL of SYBR Green Real-time PCR Master Mix, 0.8 µL of 10 pmol/μL (10 µM) each primer, 2 µL of template cDNA, and PCR grade water. Real-time qRT-PCR conditions were 95 °C for 10 min, 40 cycles of 95 °C for 15 s, 56 °C for 10 s  and 72 °C for 15 s, and by a melting curve stage of 95 °C for 10 s and 60 °C for 1 min. Actin primers were used for each sample as the internal positive control. Relative gene expression levels were calculated using the 2^−ΔΔCT^ method.

### 2.15. Gas Chromatography-Mass Spectrometry (GC-MS) Analysis

AY001 was inoculation in 1 L of NB medium for 2 days. After cultivation, it was centrifuged at 10,000× *g* for 10 min and supernatant transferred to the conical flasks. In each conical flask, hexane or ethyl acetate was added to supernatant in a 1:1 ratio, shaken, and kept overnight. Then, extraction solution was dried using a rotary evaporator. Each extract was weighed and dissolved either in methanol for GC-MS analysis or in DMSO for antifungal activity test. GC/MSD System (5977A Series, Agilent Technologies, Santa Clara, CA, USA) was used. The oven temperature was held at 40 °C for 1 min and then increased to 300 °C at 10 °C/min rate. The injector and mass interface temperature was 300 °C. The carrier gas was helium at a flow rate of 1 mL/min. Injection mode was split, and mass range from 50 to 400 *m/z* was scanned. The result was analyzed using the National Institute of Standards and Technology spectral library version 11 (NIST 11 spectral library).

## 3. Results

### 3.1. Isolation and Identification of Burkholderia Conataminans AY001

Five different soil samples were collected and used for rhizosphere bacteria isolation. Among the 155 strains isolated and tested, AY001 showed the highest antagonistic activity against *Fusarium oxysporum* f.sp. *lycopersici* (*FOL*; Appendix A). The 16S rRNA sequence of AY001 was analyzed by using a BLAST search to find the closest neighbor genus. The results showed that AY001 was closely related to genus *Burkholderia* and AY001 showed more than 99% identity with *B. contaminans* strain JCK-CSHB12-R (accession no. MW195003.1). Phylogenetic analysis demonstrated that strain AY001 belongs to *B. contaminans* (Appendix A).

### 3.2. In Vitro Antagonistic Activity Assay

Antagonistic activity of AY001 was further tested against six different fungal pathogens, including *FOL*, *F. avenaceum*, *F. solani*, *C. acutatum*, *Phytophthora capsici* and *Sclerotinia sclerotiorum*, by dual culture assay. Although AY001 showed different levels of antagonistic activity, it was able to inhibit the growth of all the tested fungal pathogens (Figure 1A, upper panel). From the inhibition rate analysis, AY001 showed the highest antagonistic activity against *F. avenaceum* (45%) and the lowest antagonistic activity against *S. sclerotiorum* (24%) (Figure 1B). The culture filtrate of AY001 also inhibited the growth of all tested six different fungal pathogens to a different degree (Figure 1A, lower panel). Growth inhibition by culture filtrate of AY001 showed a similar trend, but its inhibition rate against *F*. *avenaceum* was lowered, but that against *C*. *acutatum* and *S. sclerotiorum* was enhanced, compared to the observation from the dual culture assay (Figure 1B,C). Culture filtrate of AY001 showed an inhibition rate of 23~43% against different fungal pathogens compared to control. Taken together, our findings suggest that AY001 has a broad-spectrum antagonistic activity against different plant pathogenic fungi.

### 3.3. Plant Growth Promoting Rhizobacteria (PGPR)-Related Traits of AY001

Different PGPR traits of AY001 were tested in vitro, (see Appendix A for the list of PGPR traits tested). AY001 successfully solubilized (or hydrolyzed) zinc oxide (ZnO), calcium phosphate (Ca_3_(PO_4_)_2_), and skim milk by forming a halo zone on tris-minimal salt agar, NBRIP and skim milk agar media, respectively (Figure 2A). Solubilization index (SI) and hydrolysis index (HI) were measured at 3, 5, 7, and 10 days after inoculation. The SI of ZnO and Ca_3_(PO_4_)_2_, and HI of skim milk were increased in a time-dependent manner (Figure 2A; Appendix A). These findings suggest that AY001 is able to solubilize insoluble zinc and phosphate, or hydrolyze the protein macromolecule into smaller molecules as well. However, AY001 did not show extracellular amylase, cellulose, and chitinase activities in vitro (Appendix A).

Other PGPR-related traits, such as siderophore, ammonia and IAA production, and nitrogen fixation activities, were also tested. Siderophore production activity was tested until 5 days on CAS media. AY001 produced a halo zone on CAS media, suggesting it is able to produce a siderophore (Figure 2B). Ammonia production by AY001 was measured at 5, 7, and 10 days after inoculation in peptone broth media (Figure 2C). After mixing the AY001 culture with Nessler’s reagent, it developed orange color, suggesting successful ammonia production at 5, 7, and 10 days after inoculation (Figure 2C; color changes at 0 and 10 days after inoculation are shown in the inlet). The highest amount of ammonia production was observed at 10 days after inoculation. IAA production of AY001 was tested in NB media with different concentrations of L-tryptophan. AY001 was able to produce IAA in the presence of L-tryptophan in a dose-dependent manner (Figure 2D). Nitrogen fixation activity of AY001 was tested in an N_2_ free BAz medium by using azelaic acid as a carbon source (Estrada et al. 2001). The color of the BAz medium changed from yellow to blue after 7 days (Figure 2E; media color change is shown in the inlet). Nitrogen fixation activity of AY001 was also significantly increased as observed by increased pH from 5.7 to 8.67 after 7 days (Figure 2E). Taken all the available evidence together, AY001 showed various PGPR-related traits in vitro.

### 3.4. PGPR and Biocontrol Activities of AY001 in Tomato Plants

The ability of PGPR to colonize the root is essential for its effective PGPR activity. Thus colonization of AY001 was tested by quantitative measurement of the bacterial population in tomato roots. The population of AY001 in tomato roots was monitored up to 14 days post-inoculation by (dpi; Figure 3). The population of AY001 was increased up to 3 × 10^7^ cfu/g at 14 dpi, suggesting it is able to colonize the tomato root system.

As AY001 showed various PGPR-related traits in vitro and successful colonization in tomato roots, its growth-promoting effect is examined (Figure 3C–F). Treatment of tomato roots with a low concentration of AY001 (2 × 10^5^ cfu/mL) significantly increased the fresh and dry weight of roots, but not of shoot compared to water-treated control; however, treatment with high concentration of AY001 (2 × 10^7^ cfu/mL) significantly enhanced fresh and dry weight of both root and shoot. In plants treated with a high concentration of AY001 (2 × 10^7^ cfu/mL), root weight was improved by 26% (fresh weight) and 67% (dry weight), and shoot weight was improved by 52% (fresh weight) and 77% (dry weight). This suggests that AY001 is able to act as an effective PGPR in tomato plants.

As AY001 showed antagonistic activity against various plant pathogenic fungi (Figure 1), its biocontrol activity in tomato plants was examined by using the *FOL*, a causal agent of tomato *Fusarium* wilt disease. Tomato plants inoculated with *FOL* showed very severe growth retardation and wilting phenotype at 2 weeks after inoculation (Figure 4A); however, pre-treatment of tomato roots with AY001 significantly recovered plant growth parameters and reduced disease symptoms (Figure 4A,B). In particular, high titer of AY001 (2 × 10^7^ cfu/mL) showed significantly higher protective effect, as it showed significantly higher length of shoot and root, and fresh and dry weights of shoot and root, compared to mock- or low titer of AY001 (2 × 10^5^ cfu/mL)-treated ones (Figure 4B–F). Taken all the available evidence together, AY001 is not only able to promote the growth of tomato plants, but also protect tomato plants from *FOL* by acting as a root-colonizing PGPR.

### 3.5. ISR-Inducing Activity of AY001 in Tomato Plants

To further analyze whether AY001 can protect tomato plants against different pathogens, we used *Pst* DC3000, a causal agent of bacterial speck disease. As shown in Figure 5A, *Pst* DC3000 induced the typical symptoms of bacterial speck disease, such as brown spots and extensive chlorosis encirclement, on the inoculated leaves; however, *Pst* DC3000-induced disease symptoms were reduced on the tomato plants pre-treated with AY001 on their roots or leaves at 24 h before *Pst* DC3000 inoculation (Figure 5A). Pretreatment on both tomato roots (AY001_R_) and leaves (AY001_L_) with AY001 significantly reduced bacterial growth of *Pst* DC3000 (Figure 5B).

To examine whether root treatment of tomato plants with AY001 can trigger Induced Systemic Resistance (ISR), RT-PCR and real-time qRT-PCR were performed to analyze the expression of jasmonic acid/ethylene (JA/ET) signaling pathway-related marker genes, *PIN2*, *LapA,* and *ACO1*. In both analyses, expression levels of *PIN2*, *LapA*, and *ACO1* genes were distinctly up-regulated in the leaves after 12 and 18 hpi (Figure 5C–F). In particular, the expression level of the *ACO1* gene showed the highest expression levels at 18 hpi. This suggests that AY001 not only enhances the growth of tomato plants through its PGPR activity, but also protects tomato plants against pathogen infection by triggering ISR.

### 3.6. Gas Chromatography-Mass Spectrometry (GC-MS) Analysis

To analyze secondary metabolites in culture filtrate of AY001, hexane and ethyl acetate were used as extraction solvents. GC-MS analysis of each fraction showed different profiles of compounds. Thirty-two different peaks were observed from hexane extract, while 61 different peaks observed from ethyl acetate extract. The total ion chromatograph (TIC) corresponding to the compounds extracted with hexane and ethyl acetate from culture filtrate of AY001 was shown in Figure 6. In each extract, the top five most abundant chemical compounds were listed separately (Table 1 and Table 2).

## 4. Discussion

*Burkholderia cepacia* complex (Bcc) is found naturally in soil, water, and rhizosphere of plants, and some of them are known to exhibit various PGPR-related traits, including phosphate and zinc solubilization, siderophore and IAA formation, and nitrogen fixation activities [27,30,31,32,33,34]. In this study, we newly isolated and identified a *B*. *contaminans* AY001 with distinct PGPR and biocontrol activities. AY001 exhibited various PGPR-related traits in vitro, and successfully colonized tomato roots and enhanced the growth of tomato plants. When compared to the non-treated controls, 2 × 10^7^ cfu/mL AY001-treated tomato plants showed 52% and 77% greater shoot fresh and dry weights, respectively. Similarly, 26% and 67% greater root fresh and dry weights were observed in 2 × 10^7^ cfu/mL AY001-treated plants compared to non-treated control plants. Interestingly, different members of (Bcc) are also known to be able to successfully control different diseases in different host plants [30,35,36]. For example, *B*. *contaminans* KNU17BI1 showed biocontrol activity against banded leaf and sheath blight of maize seedling caused by *R*. *solani* [33]. *Burkholderia cepacia JBK9* and its n-hexane-extracted fraction showed distinct antifungal activity against different plant pathogens, including *Phytophthora capsici*, *Fusarium oxysporum,* and *Rhizoctonia solani*, and indeed suppressed Phytophthora blight of red pepper plants [37]. To our knowledge, this is the first report on the biocontrol of tomato Fusarium wilt disease by using *B*. *contaminas*. Together, AY001 showed great potential as both bio fertiliser and biocontrol agent of tomato plants. Furthermore, the level of colonization of AY001 in tomato roots reached 3 × 10^7^ cfu/g at 14 dpi. Although AY001 was able to maintain high population numbers in tomato roots, we did not observe any disease-like symptoms in tomato plants treated with AY001. Its high capacity to colonize the tomato roots may facilitate stable PGPR and biocontrol activities in tomato plants over time.

The ISR-inducing activity of AY001 in tomato plants was also determined by testing the expression of JA/ET-pathway marker genes and defense response against *Pst* DC3000. AY001 did not show any direct antibacterial activity against *Pst* DC3000 in vitro. However, root treatment of AY001 enhanced the expression of *proteinase inhibitor II* (*PIN2*), *leucine aminopeptidase A* (*LapA*), and *1-aminocyclopropane-1-carboxylate oxidase 1* (*ACO1*) genes, and significantly reduced bacterial growth of *Pst* DC3000 in tomato plants. In plants, two different forms of resistance responses, systemic acquired resistance (SAR) and Induced Systemic Resistance (ISR), are known to be induced by different elicitors [38,39,40]. Unlike SAR is induced by avirulent pathogen infection and salicylic acid (SA)-dependent pathway, ISR is known to be induced by PGPR and JA/ET-dependent pathways. In tomato plants, expression of JA-responsive *PIN2* is strongly induced during the wounding stress and arbuscular mycorrhizal fungi-mediated enhanced resistance to early blight caused by *Alternaria solani* [41,42]. Another JA-responsive gene *LapA* is a positive regulator of late wound responses and its expression is induced during the ISR induction by *Bacillus amyloliquefaciens* MBI600 [43,44]. ACO1 is involved in the biosynthesis of ET via converting 1-aminocyclopropane-1-carboxylic acid into ET [45]. Thus enhanced expression of *ACO1* in AY001 treated plants suggest enhanced ET signaling. Taken together, AY001 not only induced plant growth promotion and resistance to Fusarium wilt disease via direct antifungal activity, but also induced ISR, thereby enhancing the resistance to bacterial pathogen *Pst* DC3000.

There is growing interest in the use of secondary metabolites of *Burkholderia* spp. in agriculture [19,46,47]. Notably, comparative genome analysis of *B*. *contaminans* MS14 with other 17 *Burkholderia* spp. revealed that the presence of multiple genes is related with antimicrobial secondary metabolite production, but lesser genes contribute to pathogenicity and virulence in plants and animals [19]. In this study, GC-MS analysis of hexane- and ethyl acetate-extract of culture filtrate of AY001 identified multiple and distinct secondary metabolites, including (1) di(2-ethylhexyl) phthalate; (2) octadec-9-enoic acid; (3) 2-ethyl-1-hexanol; (4) (Z)-9-octadecenamide; (5) hexadecane; (6) pyrrolo[1,2-a]pyrazine-1,4-dione, hexahydro-3-(phenylmethyl); (7) pyrrolo[1,2-a]pyrazine-1,4-dione, hexahydro-3-(2-methylpropyl)-; (8) L-proline, N-pivaloyl-, ethyl ester; (9) 2,4(1H,3H)-pyrimidinedione, 1,3,6-trimethyl-; and (10) N-benzyl-N-ethyl-p-isopropylbenzamide. (1) Di(2-ethylhexyl) phthalate (DEHP) is the most commonly used phthalate as plasticizers in polyvinyl chloride (PVC) and other polymers to enhance its durability and elasticity [48]. There are conflicting interests of DEHP on human and plant health [49,50]. Although acute toxicity of DEHP is relatively low [LD_50_ = 30 g/kg in rats (oral)], DEHP is regarded as pollutants due to its extensive use as a plasticizer, and its potential as an endocrine disruptor [50]. On the other hand, DEHP is found in medicinal plants *Calotropis gigantea* L. (Asclepiadaceae), and showed antibacterial and antifungal activities against different microorganisms [49]. Thus the use of DEHP for plant protection needs to be considered with caution. (2) Octadec-9-enoic acid (also known as oleic acid) is an omega-9 fatty acid, which is known to have several biological activities, such as antibacterial activity against gram-positive bacteria [51] and plant immune-inducing activities [52]. (3) 2-Ethyl-1-hexanol (2EH) is widely used as solvents, flavors, fragrances, and precursors of DEHP production. In a recent study, rhizosphere bacteria of rice plants produced 2EH as a volatile organic compound and it showed antifungal activity against rice sheath blight pathogen, *Rhizoctonia solani* [53]. (4) (Z)-9-octadecenamide is known to be produced by *B*. *contaminans* NZ, which reportedly exhibit tyrosinase inhibitory, antifungal, and antibiotic activities [54]. (5) In Arabidopsis plants, hexadecane treatment-induced ISR against two different bacterial pathogens, *P*. *syringae* pv. *maculicola* and *Pectobacterium carotovorum* [55]. (6) Pyrrolo[1,2-a]pyrazine-1,4-dione, hexahydro-3-(phenylmethyl)- (PPDHP) is reportedly identified from *Streptomyces* sp. VITPK9 and showed antifungal activity with low cytotoxicity [56]. (7) Pyrrolo[1,2-a]pyrazine-1,4-dione, hexahydro-3-(2-methylpropyl)-, which has a similar structure to PPDHP, identified from the ethyl acetate extract of *Fusarium* sp., and showed antibacterial activity against both gram-positive and gram-negative bacteria [57]. Regardless of rigorous search on the biological significance and/or source of (8) L-proline, N-pivaloyl-, ethyl ester; and (9) 2,4(1H,3H)-Pyrimidinedione, 1,3,6-trimethyl-, unfortunately, we could not find significant information. Finally, (10) N-benzyl-N-ethyl-p-isopropylbenzamide is identified as one of 22 bioactive compounds from antimicrobial methanol extract of *Klebsiella pneumonia* [58]. However, its antimicrobial activity is not solely tested yet. Taken together, hexane- and ethyl acetate-extract of culture filtrate of AY001 identified different profiles of secondary metabolites with direct antimicrobial or with ISR inducing activities. Biocontrol is a promising strategy to reduce the loss of plant production in agriculture. Our study established that potential use of AY001 as a biocontrol agent and/or biofertilizer in tomato cultivation. Further safety and formulation studies and field experiments will be needed to ensure the successful and practical use of AY001 in tomato cultivation.

## 5. Conclusions

A novel Burkholderia *contaminans* AY001 exhibited various PGPR traits, such as nitrogen fixation, phosphate solubilization, extracellular protease production, zinc solubilization and IAA biosynthesis activities. Its treatment not only enhanced the growth of tomato plants, but also enhanced resistance to *Fusarium oxysporum* f. sp. *lycopersici* and *Pseudomonas syringae* pv. *tomato* DC3000. GC-MS analysis of culture filtrate of AY001 identified distinct secondary metabolites with antimicrobial activities.

## Figures and Tables

**Figure 1 biology-11-00619-f001:**
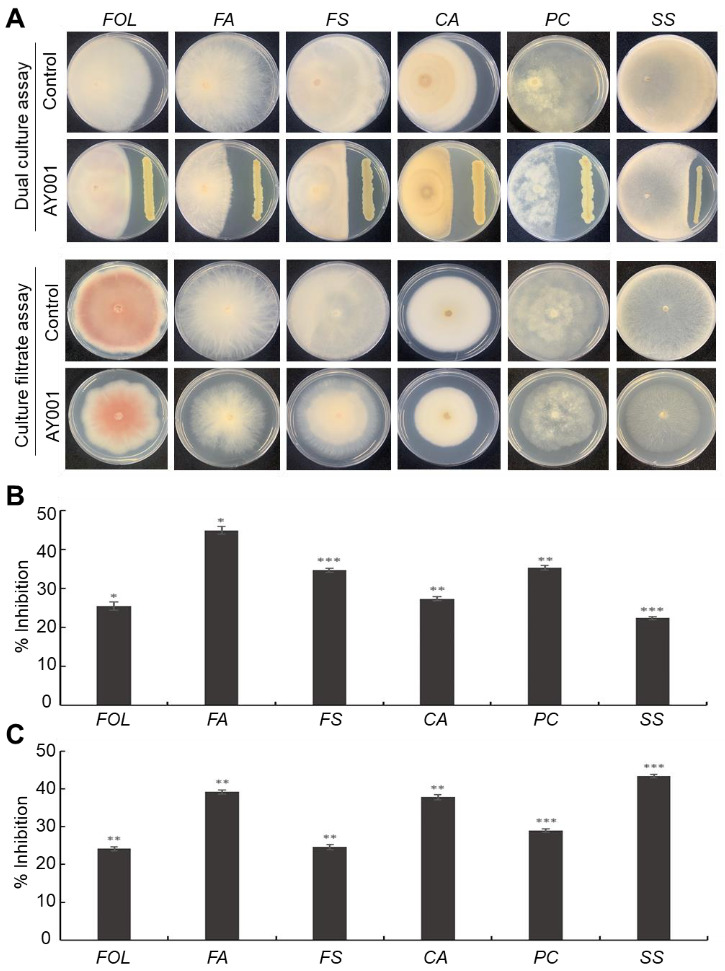
Anagonistic activity of AY001 against different plant pathogenic fungi. (**A**) Representative picture of dual culture (upper panel) and culture filtrate (lower panel) assays of AY001 against *Fusarium oxysporum* (*FOL*), *F. avenaceum* (*FA*), *F. solani* (*FS*), *C. acutatum* (*CA*), *Phytophthora capsici* (*PC*)and *Sclerotinia sclerotiorum* (*SS*). Control: pathogen growth without AY001. AY001: pathogen growth with AY001. (**B**,**C**) Inhibition rate of AY001 against different plant pathogenic fungi in dual culture (**B**) and culture filtrate (**C**) assays at up to 14 days after incubation (* *p* < 0.05; ** *p* < 0.005; *** *p* < 0.0005).

**Figure 2 biology-11-00619-f002:**
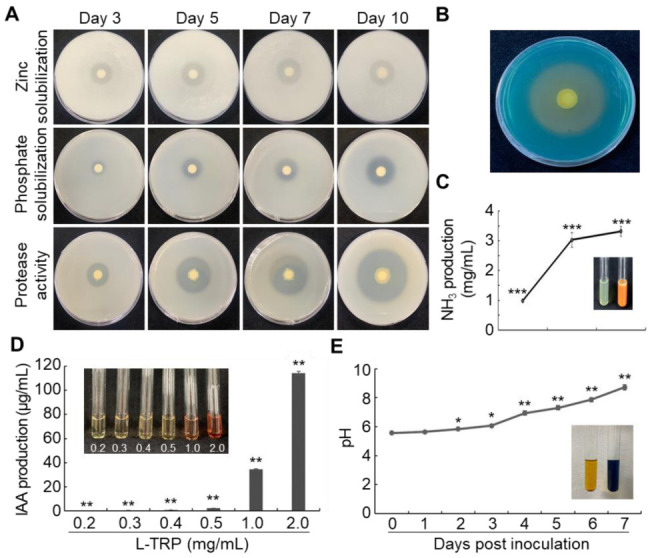
Plant-growth promotion (PGP)-related traits of AY001. (**A**) Extracellular zinc solubilization, phosphate solubilization, and protease activities of AY001 in tris-minimal salt agar, National Botanical Research Institute’s phosphate (NBRIP) and skim milk agar media, respectively. AY001 was grown on indicated media for up to 10 days at 25 °C. (**B**–**D**) Siderophore (**B**), ammonia (**C**), and IAA (**D**) production activities of AY001 on Chrome azurol S (CAS), peptone broth, and nutrient broth media, respectively. (**E**) Nitrogen fixation activity of AY001. AY001 was incubated in NB medium at 28 °C in the presence of indicated concentrations of L-tryptophan. Data are mean ± standard deviation. Asterisks indicate a significant difference (* *p* < 0.05; ** *p* < 0.005; *** *p* < 0.0005).

**Figure 3 biology-11-00619-f003:**
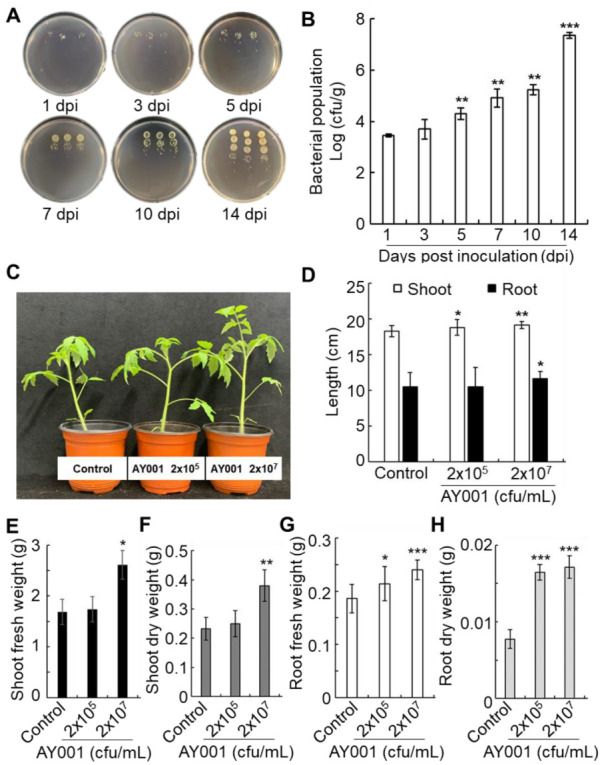
PGPR activity of AY001 in tomato plants. (**A**) Root colonization of AY001 was analyzed by dot-plating. (**B**) AY001 population levels at 1, 3, 5, 7, 10 and 14 dpi. (**C**) Enhanced growth phenotype of tomato plants at 2 weeks after treatment with different concentrations of AY001. (**D**–**H**) Different plant growth parameters measured at 2 weeks after treatment with different concentrations of AY001. (**D**) Shoot and root length. (**E**) Shoot fresh weight. (**F**) Shoot dry weight. (**G**) Root fresh weight. (**H**) Root dry weight. Data are mean ± standard deviation. Asterisks indicate a significant difference (* *p* < 0.05; ** *p* < 0.005; *** *p* < 0.0005).

**Figure 4 biology-11-00619-f004:**
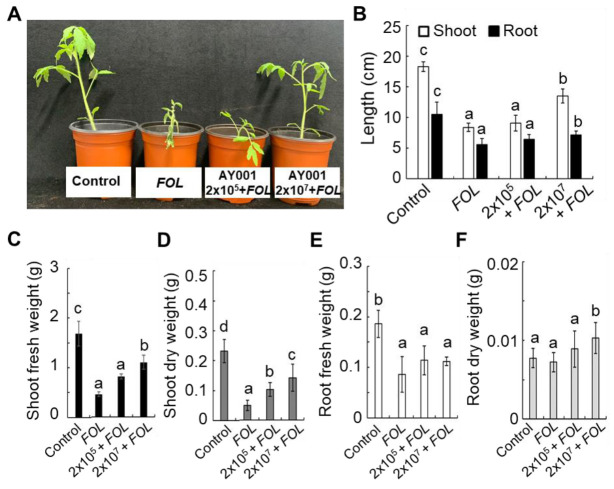
Biocontrol activity of AY001 against *FOL*. (**A**) Enhanced resistance of tomato plants against *FOL* by AY001. Two-week-old tomato plants were treated with different concentrations of AY001 by root dipping methods at 1 day before inoculation with *FOL*. Pictures were taken 2 weeks after inoculation. (**B**–**F**) Enhanced resistance of tomato plants against *FOL* by AY001 treatment. Different plant growth parameters measured at 2 weeks after treatment with different concentrations of AY001. (**B**) Shoot and root length. (**C**) Shoot fresh weight. (**D**) Shoot dry weight. (**E**) Root fresh weight. (**F**) Root dry weight. Data are mean ± standard deviation. Different letters indicate a significant differences between treatments (*p* < 0.05).

**Figure 5 biology-11-00619-f005:**
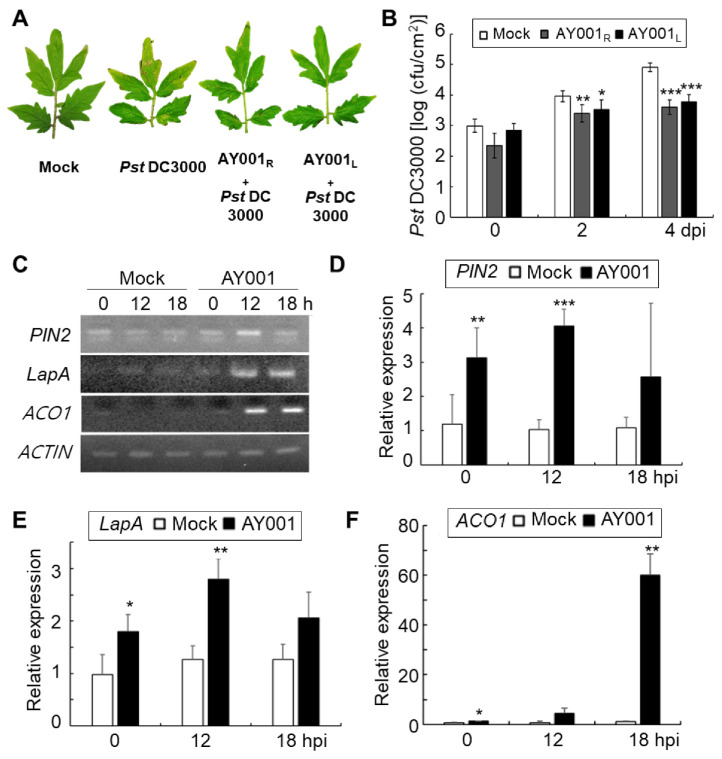
The ISR-inducing activity of AY001 in tomato plants. (**A**) Biocontrol activity of AY001 against bacterial speck disease of tomato. Three-week-old tomato plants were treated with AY001 either by root dipping (AY001_R_) or spray (AY001_L_) methods at 1 day before inoculation with *Pst* DC3000. Pictures were taken at 4 dpi. (**B**) Bacterial growth in tomato leaves treated with AY001. (**C**) RT-PCR analysis of the expression of *PIN2*, *LapA,* and *ACO1*. The level of Actin was visualized as a control. (**D**–**F**) RT-qPCR analysis of *PIN2* (**D**), *LapA* (**E**), and *ACO1* (**F**) expressions. Data are mean ± standard deviation. Asterisks indicate a significant difference (* *p* < 0.05; ** *p* < 0.005; *** *p* < 0.0005).

**Figure 6 biology-11-00619-f006:**
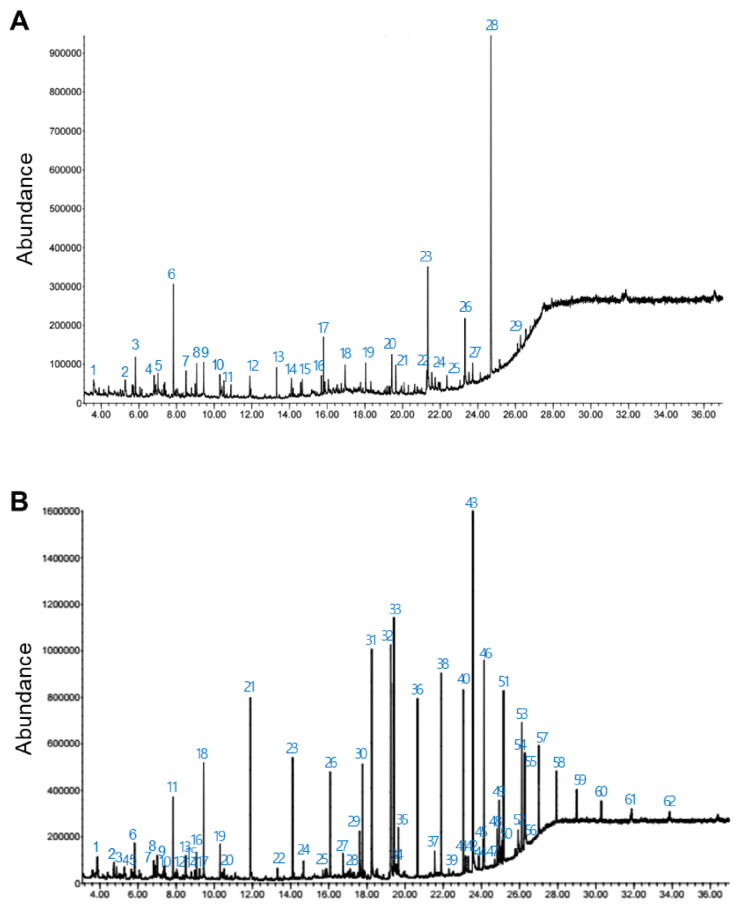
GC-MS total ion chromatograms (TICs) obtained from the analysis of culture filtrate of AY001. (**A**) TICs obtained from the analysis of hexane extract, (**B**) TICs obtained from the analysis of ethyl acetate extract. The numbers refer to the substances in Table 1 and Table 2.

**Table 1 biology-11-00619-t001:** List of five of the most abundant chemical compounds of hexane extract by GC-MS analysis.

Peak No.	Compound	Structure	Formula	Molecular Weight	Retention Time (min)
28	Di(2-ethylhexyl) phthalate	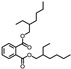	C_24_H_38_O_4_	390.277	24.695
23	Octadec-9-enoic acid	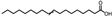	C_18_H_34_O_2_	282.256	21.344
6	2-Ethyl-1-hexanol	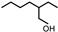	C_8_H_18_O	130.136	7.834
26	(Z)-9-Octadecenamide	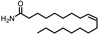	C_18_H_35_NO	281.272	23.318
17	Hexadecane	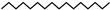	C_16_H_34_	226.266	15.807

**Table 2 biology-11-00619-t002:** List of five of the most abundant chemical compounds of Ethyl acetate extract by GC-MS analysis.

Peak No.	Compound	Structure	Formula	Molecular Weight	Retention Time (min)
43	Pyrrolo[1,2-a]pyrazine-1,4-dione, hexahydro-3-(phenylmethyl)	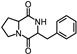	C_14_H_16_N_2_O_2_	244.121	23.552
33	Pyrrolo[1,2-a]pyrazine-1,4-dione, hexahydro-3-(2-methylpropyl)-	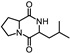	C_11_H_18_N_2_O_2_	210.137	19.418
32	L-Proline, *N*-pivaloyl-, ethyl ester	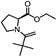	C_12_H_21_NO_3_	227.152	19.25
31	2,4(1*H*,3*H*)-Pyrimidinedione,1,3,6-trimethyl-	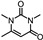	C_7_H_10_N_2_O_2_	154.074	18.255
21	*N*-Benzyl-*N*-ethyl-p-isopropylbenzamide	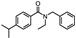	C_19_H_23_NO	281.178	14.102

## Data Availability

Not applicable.

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
