# Peer review of "Biological Control Activity of Plant Growth Promoting Rhizobacteria Burkholderia contaminans AY001 against Tomato Fusarium Wilt and Bacterial Speck Diseases"

_biology, 2022, doi:10.3390/biology11040619_

Round 1

Reviewer 1 Report

After carefully assessing the article, below are my comments:

1. Introduction

In the introduction, please add information on the impact of the proposed biological factors on humans and animals, because it is important for the practical application of future biopreparations

2. Materials and Methods
2.11. AY001 treatment and FOL inoculation

Please indicate the composition of the substrate used for the pots and whether the pots have been previously sterilized.

Supplementary Figure S1. AND
Please enlarge part B as it is hardly legible.

Yours sincerely

Author Response

Reviewer 1

After carefully assessing the article, below are my comments:

  1. Introduction

In the introduction, please add information on the impact of the proposed biological factors on humans and animals, because it is important for the practical application of future biopreparations

Response: Thank you very much for constructive review. As reviewer suggested, we have included safety concern about using AY001 for further usage as biological control agent in “simple summary (lines 16-17)” and “introduction (lines 66-75)” sections. 

  1. Materials and Methods
    2.11. AY001 treatment and FOL inoculation

Please indicate the composition of the substrate used for the pots and whether the pots have been previously sterilized.

Response: As reviewer indicated, we have provided the substrate information (Lines 204-206). For your information, we have directly used substrate without autoclave.

Supplementary Figure S1. AND
Please enlarge part B as it is hardly legible.

Response: As reviewer suggested, we have enlarged Figure S1B and provided as separate file submission system.  

Reviewer 2 Report

A Yeong Heo et al's manuscript "Biological control activity of plant growth-promoting rhizobacteria 2 Burkholderia contaminans AY001 against tomato Fusarium wilt and bacterial speck diseases" is well written. Extensive experiments have been applied to examine the PGPR-related traits and antagonistic and ISR-inducing activities. Overall, I did not notice any obvious errors in this manuscript. Only a few points need to be clarified before the acceptance. 

  1. Line 9, The Burkholderia cepacia complex (B. cepacia) consists of different species of bacteria that are found in the natural environment. Some of these species pose serious risks to the health of a person with cystic fibrosis. It would be more convenient if the authors could add their opinions on the safety concern of the environmental application of the isolate.
  2. Line 20, the anti-antibiotic ability should be also investigated.
  3. Line 26, what is "JA/ET"? 
  4. Line 87, any purification process for the isolate?
  5.  The authors should detail the recipes of all media mentioned in the text.
  6. There is a typo in Line 354 "Supplementary Table S3. Qualitative analysis of znic and phosphate solubilization and protease efficiency of AY001 at 10 dai (Mean ± SD)." It should be "zinc" rather than "znic" .

Author Response

A Yeong Heo et al's manuscript "Biological control activity of plant growth-promoting rhizobacteria 2 Burkholderia contaminans AY001 against tomato Fusarium wilt and bacterial speck diseases" is well written. Extensive experiments have been applied to examine the PGPR-related traits and antagonistic and ISR-inducing activities. Overall, I did not notice any obvious errors in this manuscript. Only a few points need to be clarified before the acceptance. 

1. Line 9, The Burkholderia cepacia complex (B. cepacia) consists of different species of bacteria that are found in the natural environment. Some of these species pose serious risks to the health of a person with cystic fibrosis. It would be more convenient if the authors could add their opinions on the safety concern of the environmental application of the isolate.

Response: Thank you very much for constructive review. As reviewer suggested, we have included safety concern about using AY001 for further usage as biological control agent in “simple summary (lines 16-17)” and “introduction (lines 66-75)” sections.

2. Line 20, the anti-antibiotic ability should be also investigated.

Response: In this study, we have only found anti-fungal activity of AY001 against different phytopathogenic fungi, including FOL, but not against bacterial pathogen Pseudomonas syringae pv. tomato DC3000 (Described in Line 513). We believe that enhanced resistance of AY001-treated tomato plants against Pst DC3000 is likely dependent on AY001’s ISR inducing activity rather than direct antibacterial activity. Thus we could not include antibiotic ability of AY001 in abstract section.

3. Line 26, what is "JA/ET"? 

Response: As suggested, we spelled it out into jasmonic acid/ethylene (Line 27; Line 461)

4. Line 87, any purification process for the isolate?

Response: As described in materials in methods section, gDNA was isolated according to the manufacturer’s instruction. Thus, we may not describe the details of gDNA extraction protocol, which is commonly provided together with commercial gDNA isolation kit.

5. The authors should detail the recipes of all media mentioned in the text.

Response: I have double checked the materials and methods section, and found that we have provided recipes of all media used.

6. There is a typo in Line 354 "Supplementary Table S3. Qualitative analysis of znic and phosphate solubilization and protease efficiency of AY001 at 10 dai (Mean ± SD)." It should be "zinc" rather than "znic".

Response: Thank you very much. We have corrected typo errors as reviewer suggested (Supplementary Table S3 legend, Line 578).